# Cellular Localization of Orexin 1 Receptor in Human Hypothalamus and Morphological Analysis of Neurons Expressing the Receptor

**DOI:** 10.3390/biom13040592

**Published:** 2023-03-25

**Authors:** Konstantina Vraka, Dimitrios Mytilinaios, Andreas P. Katsenos, Anastasios Serbis, Stavros Baloyiannis, Stefanos Bellos, Yannis V. Simos, Nikolaos P. Tzavellas, Spyridon Konitsiotis, Patra Vezyraki, Dimitrios Peschos, Konstantinos I. Tsamis

**Affiliations:** 1Department of Physiology, Faculty of Medicine, School of Health Sciences, University of Ioannina, 45110 Ioannina, Greece; 2Kenhub GmbH, 04318 Leipzig, Germany; 3Department of Pediatrics, University Hospital of Ioannina, 45500 Ioannina, Greece; 4Faculty of Medicine, School of Health Sciences, Aristotle University, 54124 Thessaloniki, Greece; 5Department of Neurology, University Hospital of Ioannina, University of Ioannina, 45500 Ioannina, Greece

**Keywords:** orexin receptor 1, hypothalamus, functional neuroanatomy, orexin, Golgi staining

## Abstract

The orexin system is related to food behavior, energy balance, wakefulness and the reward system. It consists of the neuropeptides orexin A and B, and their receptors, orexin 1 receptor (OX1R) and orexin 2 receptor (OX2R). OX1R has selective affinity for orexin A, and is implicated in multiple functions, such as reward, emotions, and autonomic regulation. This study provides information about the OX1R distribution in human hypothalamus. The human hypothalamus, despite its small size, demonstrates a remarkable complexity in terms of cell populations and cellular morphology. Numerous studies have focused on various neurotransmitters and neuropeptides in the hypothalamus, both in animals and humans, however, there is limited experimental data on the morphological characteristics of neurons. The immunohistochemical analysis of the human hypothalamus revealed that OX1R is mainly found in the lateral hypothalamic area, the lateral preoptic nucleus, the supraoptic nucleus, the dorsomedial nucleus, the ventromedial nucleus, and the paraventricular nucleus. The rest of the hypothalamic nuclei do not express the receptor, except for a very low number of neurons in the mammillary bodies. After identifying the nuclei and neuronal groups that were immunopositive for OX1R, a morphological and morphometric analysis of those neurons was conducted using the Golgi method. The analysis revealed that the neurons in the lateral hypothalamic area were uniform in terms of their morphological characteristics, often forming small groups of three to four neurons. A high proportion of neurons in this area (over 80%) expressed the OX1R, with particularly high expression in the lateral tuberal nucleus (over 95% of neurons). These results were analyzed, and shown to represent, at the cellular level, the distribution of OX1R, and we discuss the regulatory role of orexin A in the intra-hypothalamic areas, such as its special role in the plasticity of neurons, as well as in neuronal networks of the human hypothalamus.

## 1. Introduction

Orexin A and B are neuropeptides consisting of 33 and 28 amino acids, respectively, that were discovered in 1998 by two independent research teams [1,2]. Orexin-producing neurons are located in the perifornical nucleus, in the dorsal and lateral hypothalamic areas, [3] while their axons project widely throughout the central nervous system (CNS), except for the cerebellum [4,5]. Specifically, the orexinergic neurons mainly project to nuclei in the brain stem and hypothalamus, such as the ventral tegmental area, locus coeruleus, tuberomammillary nucleus, raphe nuclei, and laterodorsal tegmental nuclei [4,5]. The orexins are ligands for two G protein-coupled receptors, namely orexin 1 (OX1R) and orexin 2 receptors (OX2R), whose amino acid sequences are 64% similar to each other in humans. The OX1R has selective affinity for orexin-A, and the OX2R has almost equal binding affinity for both orexins [1]. OX1R and OX2R consist of seven transmembrane helices, of which the C-terminal area, comprising two to four helices, is responsible for the difference in binding affinity [6].

The orexin system has been shown to be involved in food-seeking behavior, energy balance, wakefulness and the reward system [7]. OX2R is mainly related to sleep–wake control, while OX1R is implicated in a multitude of functions, such as reward, emotion, and autonomic regulation [8]. Deficiency of the orexin system due to a loss of orexin-producing neurons is associated with narcolepsy, a disease characterized by excessive daytime sleepiness. Since the orexin system is involved in many physiological and pathological processes, its receptors are considered to be potential therapeutic targets, and both antagonists and agonists are being developed. Suvorexant and Lemborexant are two dual orexin receptor antagonists, which were approved by the FDA for insomnia treatment in 2014 and 2019, respectively [9].

The hypothalamus, despite being a small part of the human brain, is implicated in many important functions, such as thermoregulation, fluid and electrolytes balance, sex behavior and reproduction, energy metabolism and feeding, sleep–wake cycles and endocrine control. It is located below the thalamus, from which it is separated by the hypothalamic sulcus, and lies anterior to the mammillary bodies, posterior to optic chiasm and between the optic tracts [10]. Cytoarchitecturally, it is divided into three regions, namely, the anterior (preoptic and supraoptic), tuberal, and posterior regions. The anterior region includes the suprachiasmatic nucleus, the median and ventrolateral preoptic nuclei, the paraventricular nucleus, the anterior hypothalamic nucleus, and the supraoptic nucleus. The tuberal region consists of the dorsomedial, ventromedial, and arcuate nuclei. The posterior region lies above the mammillary bodies and includes them, whose nuclei form the mammillary complex, as well as the tuberomammillary and the posterior hypothalamic nuclei [11].

Studies on the rat brain have shown that OX1R and OX2R mRNAs are widely expressed in the hypothalamus and other areas of the CNS. OX1R mRNA is observed in the dorsal raphe, locus coeruleus, and tenia tecta, while OX2R mRNA is detected in the cerebral cortex, ventral tegmental area, and anterior pretectal nucleus. Both OX1R and OX2R mRNAs were found in the amygdaloid complex (OX2R only in the posterior cortical nucleus and OX1R mainly in the medial nucleus), the hippocampus, some thalamic nuclei, the subthalamic nucleus, the hypothalamus, and the pituitary gland [12,13]. There are also studies in male sheep [14], nocturnal mice [15], and wild mammals [16] that are related to mRNA of OX1R and OX2R expression in the brain and hypothalamus. In humans, studies focus mainly on the pituitary gland [17] and the hippocampus [18]. Within the rat hypothalamus, OX1R is mainly distributed within the lateroanterior hypothalamic nucleus, the ventromedial nucleus, and the dorsomedial nucleus. On the other hand, OX2R mRNA can be found in the lateral hypothalamic area, the para-ventricular nucleus, the arcuate nucleus, the ventromedial nucleus, and the dorsomedial and mammillary nuclei [12,13,19].

Except for the CNS, studies in rats show that OX1R mRNA is present in the adrenal gland, the thyroid, the jejunum, the gonads, and the kidney, and OX2R mRNA is present in the adrenal gland and the lung [20]. As for studies in humans, orexin receptors can be found in the kidney [21], the male reproductive system [22], the adrenal glands [23] and the adipose tissue [24]. Specifically, in the male reproductive system these receptors are detected in the prostate [25] and the testes [22]. In the human prostate, OX1R is located within the exocrine epithelium of the prostate [25]. In human testis Sertoli cells, testicular peritubular myoid cells and Leydig cells express both OX1R and OX2R [22]. These findings are in line with experiments in male rats [26], and support the idea that the peripheral expression of OX1R and OX2R is regulated by testosterone. In human adult adrenal glands, only OX2R can be detected in the adrenal cortex [23]. Moreover, OX1R and OX2R are detected in human adipose tissue, suggesting that orexin regulates energy balance and metabolism, not only from the CNS but also at the peripheral level [24]. Ectopic expression of OX1R can be found in colon cancer cases, regardless of the location of the primary cancer, either in the proximal or distal large intestine [27].

Further studies suggest that there are gender differences regarding orexin receptors’ expression in rats [20,28]. In the hypothalamus of female rats, there are higher levels of OX1R mRNA compared to male rats. However, in male rats there is an increased expression of OX1 receptor mRNA in the pituitary. However, neither the reason for these sex differences in the orexin system nor the functional consequences of this disparity are yet fully understood.

Due to the differences between species, it is possible that orexin receptors in the human brain are distributed in hypothalamic areas in a different way when compared to those of the rat brain. Few studies exist regarding their exact localization in the human brain [17,18], because most studies focus on peripheral tissues [21,22,23,24,25,27]. Since the orexin system and its receptors are involved in various hypothalamic processes, it is of great importance to describe, in detail, the localization of orexin receptors in the human hypothalamus, and to study the neuronal morphology of nuclei expressing the receptors. The aim of the current study was to analyze the distribution of OX1R in the human hypothalamus, thus contributing to the understanding of the orexin system’s function, as well as potentially helping in the development of new agonists and antagonists as potential treatment options for various diseases.

## 2. Materials and Methods

The study was carried out on post mortem human hypothalamus specimens from 15 individuals (6 female and 9 male), aged between 17 and 86 years, without any history of neurological or psychiatric disease, and that had a sudden death (Table 1). The most frequent cause of death was a myocardial infarction. In three instances, the cause of death was determined to be traumatic brain injury. These cases underwent a detailed macroscopic examination to ensure that the hypothalamus was not damaged. Due to the limited sample size of 15 specimens, it was challenging to make substantial conclusions from subgroup analyses based on age or gender. The primary objective of this study was to investigate the qualitative patterns of orexin receptor expression in the hypothalamus. The findings revealed that there were no significant qualitative distinctions between the groups, making it unnecessary to group the specimens based on age or gender. The study was approved by the Research Ethics Committee of the University of Ioannina, and the specimens were obtained by the Laboratory of Forensic Science and Toxicology, Faculty of Medicine. Three different methods were applied to all fifteen cases included in the study: Nissl, Golgi and immunohistochemical staining. Each hypothalamus was initially separated into three different parts, with coronal sections representing anterior, middle, and posterior aspects. From each part, a thin specimen (coronal vibratome section of 0.5 mm thickness) was removed for Nissl staining and OX1R immunohistochemistry, while the rest was used for Golgi impregnation.

The specimens that were intended for Nissl staining and immunohistochemistry were formalin-fixed for 15 days and paraffin-embedded. The Nissl method, with methylene blue, was carried out on 8 μm-thick sections for orientation purposes, observation of the cell bodies, and topographic detection of the nuclei in hypothalamus. For the rapid Golgi staining, the specimens were fixed in 10% aqueous formalin solution for at least 2 weeks, and then stained with potassium dichromate. After seven days, tissues were transferred into silver nitrate solution and stored in the dark, at room temperature, for one week. The specimens were then embedded in paraffin and were cut into 150 μm-thick sections with a Reichert microslicer (820H, Reichert-Jung Limited, Cambridge, UK). Thereafter, the specimens were deparaffined in xylene, dehydrated in graded alcoholic solutions and coverslipped directly with entelan. The Golgi-stained neurons were observed with a light microscope (Zeiss Axiostar-Plus, 1169-149, Carl Zeiss, Koln, Germany) [29].

Only a small number of neurons having all their dendrites and part of the axon were stained with the Golgi method. As such, strict criteria for the neurons’ selection were applied; only neurons that were fully impregnated, with complete dendritic arbors and not obscured by other neurons were selected for further evaluation [30]. The selected neurons (25–30 for each hypothalamic nucleus) were included in the morphological description and morphometric analysis of the hypothalamic neurons, and in the comparison between the nuclei of the hypothalamus. Then, with the help of the ImageJ program (http://rsb.info.nih.gov/ij/, accessed on 15 November 2022) the entire dendritic field was projected in two dimensions (stack). In this depiction, dendrites were mapped in a semi-automatic manner using the NeuronJ extension of the ImageJ program [31]. The Sholl method, utilizing concentric circles, was used to estimate the density of the dendritic field in relation to the distance from the cell body. According to this method, in the two-dimensional projection of the dendritic field of each neuron, using concentric circles with their radius increasing by 20 μm, were drawn, centered on the cell body. The number of dendrites crossing each concentric circle was counted.

The data for various parameters were imported into Microsoft Office Excel. For each hypothalamic nucleus, a separate dataset was generated that consisted of cell data obtained through processing with the ImageJ software, with a sample size of 25 cells per nucleus. Due to the small number of observations per nuclei, the normality of our distribution cannot be assumed (Kolmogorov–Smirnov test, *p* < 0.05 for the distribution of the length of dendrites of the cells of each nuclei), so the estimation of standard deviation or standard error for each mean would be misleading (Table 2).

A standard immunohistochemistry protocol for paraffin-embedded specimens [32] was used for the detection of OX1R in hypothalamic neurons. An antibody for OX1R (rabbit polyclonal anti-OX1R, Millipore, catalog number AB3092) was applied on 8 μm-thick paraffinized sections. This antibody is highly specific, and has been used previously in a variety of immunohistochemical studies [33,34,35]. Briefly, as a negative control, the primary antibody was omitted and immunostaining was performed, while neurons from the lateral hypothalamus were used as a positive control. For each sample, after deparaffinization, rehydration, and antigen retrieval, the slides were washed twice in TBS for 5 min and incubated with 0.3% hydrogen peroxide in methanol for 5 min to block endogenous peroxidase activity. After overnight incubation with the primary antibody in antibody dilution buffer (0.1 M TBS with 1% BSA, pH 7.4), the sections were washed twice in TBS for 5 min and incubated for 30 minutes with the peroxidase-labeled polymer (DAKO, EnVision, code K5007). Then, the sections were washed twice in TBS for 5 min and immersed for 10 min in diaminobenzidine (DAB) for chromogenic visualization, rinsed in distilled water briefly, counterstained with hematoxilin for 1 minute, dehydrated, and mounted. The sections were observed with a light microscope (Zeiss Axiostar-Plus) and were photographed at ×1000 and ×400 magnifications. The total number of neurons and the percentage of neurons in which OX1R was immunohistochemically identified were determined using ImageJ, using photomicrographs of a 32 × 10^3^ μm^2^ area that were taken randomly from different nuclei of the hypothalamus (https://imagej.nih.gov/ij/plugins/ihc-toolbox/index.html, accessed on 15 November 2022). The intensity of neuronal immunoreactivity was assessed manually. For each photomicrograph, serial pictures were captured from the same specimen’s site, in order to better distinguish cells and evaluate their immunoreactivity. Standard morphologic criteria were used for the identification of neurons, including the presence of a large cytoplasm, prominent nucleolus, nuclear envelope folding, and a patent cytoplasmic rim around the nucleus for small neurons, as well as an absence of thick rim of peripheral heterochromatin [36]. Cells without obvious neuronal characteristics were not included in the morphometric study. Figures were adjusted for brightness and contrast with the Gimp software (2.99.14, GNU Image Manipulation Program, https://www.gimp.org/, accessed on 15 November 2022).

## 3. Results

Using immunohistochemistry for the OX1R, we identified the neurons that express this receptor and the hypothalamic nuclei that those neurons belong to. The OX1R immunoreactivity on hypothalamic neurons appears to be similar to cytoplasmic granules near the cytoplasmic membrane, and is not expressed equally in all neurons. The nuclei of the human hypothalamus, where the OX1R was mainly detected, comprise the lateral hypothalamic area (lateral hypothalamic nucleus and lateral tuberal nucleus), the lateral preoptic nucleus, the supraoptic nucleus, the dorsomedial nucleus, the ventromedial nucleus, and the paraventricular nucleus. The lateral hypothalamic area and the lateral preoptic nucleus belong to the lateral zone of the hypothalamus, and this zone seems to display significant orexinergic functionality. The rest of the hypothalamic nuclei do not express the receptor, except for a very low number of neurons in the mammillary bodies. After the identification of the nuclei and the neuron groups that were immunopositive for OX1R, a morphological and morphometric analysis of those neurons was implemented using the Golgi method (Table 2).

### 3.1. Lateral Hypothalamic Area

The lateral hypothalamic area is composed of two smaller neuronal groups, the lateral hypothalamic nucleus and the lateral tuberal nucleus, and extends from the supraoptic region to the mammillary bodies. The neurons of the lateral hypothalamic area in our material were uniform in terms of their morphological characteristics, often forming small neural groups of three to four neurons. Their cell bodies were mainly bipolar or pyramidal, with a diameter typically larger than 25 μm. In this area, a very large proportion of neurons (more than 80%) expressed the OX1R, and the neurons of the lateral tuberal nucleus demonstrated a nearly uniform OX1R expression (more than 95% of the neurons). OX1R-immunopositive granules were present in the cytoplasm of the neurons and the staining intensity was substantially stronger, and this was associated with the other areas of the hypothalamus (Figure 1a). Small groups of neurons had higher immunoreactivity than isolated neurons, which had either little or no OX1R expression. 

The specific morphological and quantitative characteristics of the neuronal dendritic field were evident in the Golgi-impregnated material (Figure 1b). The quantitative analysis revealed that the mean total length of the primary dendrites was 96.72 μm, for the secondary dendrites it was 580.91 μm, and for tertiary dendrites, it was 264.1 μm. Only six neurons had quaternary dendrites, with an average length of 110 μm. The number of dendritic segments in each neuron was mostly stable, with an average of 14. Sholl analysis of the neurons in the lateral hypothalamic area showed that the dendrites display a higher density at a distance of 60–80 μm away from the cell body.

### 3.2. Lateral Preoptic Nucleus

The lateral preoptic nucleus is located in the lateral and anterior region of the hypothalamus. It is a small, oval-shaped nucleus that contains a diverse population of neurons. The neurons of this area can vary in size, but they are generally small to medium, with cell bodies ranging from about 10 to 30 μm in diameter. The shape of the cell body may be oval or polyhedral. Almost all neurons (more than 95%) expressed OX1,R but with low intensity of the immunoreactivity (Figure 2a). The OX1R appeared as cytoplasmic granules clustered near the cell membrane.

The neurons of the lateral preoptic nucleus appeared to be complex, and the dendrites extended up to 300 μm away from the cell body (Figure 2b). Even quinary dendrites could be found in some neurons. Their axons were usually directed towards the lateral hypothalamic area. Each neuron had three to eight primary dendrites, which were wide but short. Close to the cell body (15–40 μm), the primary dendrites branched into secondary dendrites. The neurons had, on average, about six secondary dendrites, with a mean total length 693.21 μm. The mean number of tertiary dendrites per neuron was also high (about six) and their length varied, with a mean of 156.3 μm. Quaternary dendrites were observed in 17 neurons, which had a mean total length of 64.9 μm. The total length of the dendritic field in these neurons varied from 1500 to 2500 μm, with an average length of 1893 μm. Sholl analysis of the neurons indicated that the dendrites displayed higher densities at a distance of 160 μm from the cell body.

### 3.3. Dorsomedial Nucleus

The dorsomedial nucleus is divided into two compartments: the compact compartment and the diffuse compartment. In our material, the compact compartment appeared in Nissl-stained sections, with a high density of neurons, either bipolar or ovoid, and with the diameters of the cell bodies ranging between 15 and 25 μm. Regarding the presence of the OX1R, only neurons located in the compact compartment expressed OX1R (Figure 3a). More than 90% of those neurons had dense cytosolic immunopositivity in OX1R.

After Golgi staining, these neurons appeared to have a small number of primary dendrites (2–3), extending far from the body with sparse branching (Figure 3b). There were just a few secondary and tertiary dendrites on these neurons, while the axons were directed towards the lateral hypothalamic area.

### 3.4. Ventromedial Nucleus

The ventromedial nucleus has an ovoid shape and is located near the bottom of the diencephalon, above the tuber cinereum. In the Nissl-stained material, the ventromedial nucleus could be divided into two regions: the diffuse region near the third ventricle and the compact region in the lateral part of the nucleus, consisting of two types of neurons, the main neurons (98–99%) and the small neurons (1–2%). The main neurons had a triangular cell body with a mean diameter of 10–30 μm, with a sparse dendritic tree. The small neurons were mainly located in the lateral region of the nucleus, and also had few, short dendrites. About 70% of both types of neurons were immunopositive in OX1R, and the immunoreactivity was less intense than in previously described nuclei (Figure 4a).

The main neurons had, on average, two primary dendrites, with a mean total length of 244 μm, and three secondary dendrites, with a mean total length of 502 μm. Only 5 of the 27 neurons that were observed had tertiary dendrites. Quaternary and quinary dendrites were not found in any neuron (Figure 4b). Sholl analysis of the dendrites revealed that the dendrites displayed higher densities near the cell body (40 μm).

### 3.5. Supraoptic Nucleus

The supraoptic nucleus belongs to the neuroendocrine system, and its neurons produce and excrete vasopressin and oxytocin. It is divided into three dense regions, the superolateral, the superomedial, and the inferomedial regions, that could be identified in our Nissl material, and between those regions, there is a diffuse compartment of neurons. OX1R was detected in the cytosol of more than 90% of the neurons in the dense regions (Figure 5a), while less than 50% of the neurons in the diffuse compartment had mild cytoplasmic OX1R expression. 

Evaluation of neuronal morphology after Golgi staining revealed that neurons of the dense compartments (SI neurons) were mainly pear-shaped, with large cell bodies, about 30 μm in diameter (Figure 5b). The quantitative analysis showed that each neuron had, on average, two primary dendrites, with a mean total length of 54 μm. The mean number of secondary dendrites per neuron was three, with a mean total length of 176 μm. These neurons also had, on average, four tertiary and five quaternary dendrites. These dendrites were longer than the primary and secondary ones, with average total lengths of 501 μm and 353 μm, respectively. In some neurons, even quinary dendrites were observed. Sholl analysis revealed that most dendritic segments (6-10) are located far away from the soma (140 to 180 μm). As such, the neurons of the dense compartments appeared to have some of the most complicated dendritic trees in the human hypothalamus.

The neurons of the diffuse compartment (SII neurons) are smaller with a cell body of 15 μm diameter (Figure 5c). They have a simple dendritic tree and quaternary and quinary dendrites usually do not exist. Each neuron had about 3 primary dendrites on average with a total mean length 92 μm. The number of secondary dendrites was about 3, with a total mean length 214 μm. The average number of tertiary dendrites was about 5, with a total mean length 265 μm.

### 3.6. Paraventricular Nucleus

The paraventricular nucleus is similar to the supraoptic nucleus, belonging to the neuroendocrine system. It is divided into five subnuclei, the magnocellular, the anterior parvicellular, the dorsal, the parvicellular, and the posterior nuclei [37]. The large neurons of the magnocellular compartment expressed OX1R in a proportion of 80%, while for the small neurons of the other compartments, it was 50% (Figure 6a).

Large neurons, similar to the neurons of the dense supraoptic nucleus, could be found mainly in the magnocellular and sparsely in anterior parvicellular compartments (Figure 6b). These neurons were polyhedral or pear-shaped, and their cell body had a diameter of 30 μm. The wide variation of the dendritic number, branching, and length on these neurons made the quantitative analysis of the dendritic field meaningless. Each neuron had two to three small, primary dendrites, and each primary dendrite had two to four secondary dendrites. Long tertiary, quaternary and even quinary dendrites could be found. Other neurons that were located in the paraventricular nucleus had smaller cell bodies and fewer dendrites (Figure 6c). One type of small neurons had a small, triangular cell body, and few long primary and secondary dendrites. Another type of small neuron, with cell body diameters up to 15 μm, had short primary and secondary dendrites, but quite long tertiary dendrites. The morphometric analysis is also impossible for the small neurons.

## 4. Discussion

The initial studies describing the detection of orexin receptors in the rat brain date back to 1998 [12], but this is the first study that involves the human hypothalamus, to the best of our knowledge. This study used immunohistochemistry to identify the neurons and hypothalamic nuclei that express the OX1R receptor in the human brain. The OX1R was mainly found in the lateral hypothalamic area (composed of the lateral hypothalamic nucleus and the lateral tuberal nucleus), the lateral preoptic nucleus, the supraoptic nucleus, the dorsomedial nucleus, the ventromedial nucleus, and the paraventricular nucleus. The lateral hypothalamic area and the lateral preoptic nucleus are part of the lateral zone of the hypothalamus, which appears to have significant orexinergic activity. The rest of the hypothalamic nuclei do not express the receptor, except for a very low number of neurons in the mammillary bodies. After identifying the nuclei and neuron groups that were immunopositive for OX1R, a morphological and morphometric analysis of those neurons was conducted using the Golgi method. The analysis revealed that the impregnated neurons in the lateral hypothalamic area were uniform in terms of their morphological characteristics, often forming small groups of three to four neurons. Their cell bodies were mainly bipolar or pyramidal, with a diameter usually larger than 25 μm. A high proportion of neurons in this area (over 80%) expressed OX1R, with particularly high expression in the lateral tuberal nucleus (over 95% of neurons). The neurons of the lateral preoptic nucleus appeared complex, and the dendrites extended up to 300 μm away from the cell body. Almost all neurons (over 95%) expressed OX1R, but with low intensity of immunoreactivity.

The orexin system has drawn much attention during the last decade, especially after the launching of orexin receptor antagonists in clinical practice [9,12,19,38,39,40,41]. From a developmental point of view, it is interesting that invertebrates seem to have no orexin-like sequences and in non-mammalian vertebrates, only OX2R have been identified. Regarding orexinergic neurons, the hypothalamus is the main region in which they can be found in many species among vertebrates. For example, in the chicken, orexin neurons are found in the paraventricular nucleus, the lateral hypothalamic area, and other nuclei exclusively located in the hypothalamus [39]. On the other hand, orexin neuron fibers’ locations are much more diverse among mammalian species. These fibers have been shown to project to various brain regions, such as the arcuate nucleus, median eminence, olfactory bulb, cerebral cortex, pituitary gland, area postrema, amygdala, subfornical organ, brainstem, and even to the spinal cord [40]. For example, throughout the rat brain, orexinergic axons and orexin receptors have been found to be widely distributed in several different areas [12,19]. Regarding the importance of the orexin system, the initially restricted correlation of orexin function with appetite control has gradually evolved to include sleep/wakefulness cycle regulation, and lately has been shown to have strong interactions with most functions of the autonomic and limbic systems [41,42,43]. In several species, the importance of orexins has been demonstrated for several physiological functions, such as the sleep/wake cycle in narcoleptic canines and OX2R-knockout mouse models [40]. Thus, it hasn’t come as a surprise that dual orexin receptor antagonists are related with numerous functions [9,44].

Currently, selective agonists and antagonists of orexin receptors have been tested in preclinical studies for their potential effects on insomnia, narcolepsy, obesity, addiction, anxiety, depression, dementia, and so on [9]. Especially for OX1Rs, their involvement in the regulation of autonomic responses, reward, motivation, mood, and memory further justifies and encourages their thorough investigation [45,46,47]. However, species-specific differences in the expression of orexin receptors [15] highlight the need for accurate information regarding the human brain, in order to better predict the effect of selective OX1R agonists and antagonists on patients.

The neurons of the lateral hypothalamic area have a high expression of OX1R and significant morphological and morphometric uniformity in our material. They are among the biggest neurons of the hypothalamus (together with the neurons of the paraventricular nucleus), having long dendrites with few branches and almost no dendritic spines. Dendrites with a similar morphology have been previously reported in gonadotropin-releasing hormone neurons to initiate action potentials, as part of an efficient dendritic learning procedure [48,49]. Extensive studies in experimental animals have shown that the lateral hypothalamic area is involved in addictive and reward-seeking behavior, through reciprocal connections with structures of the limbic system [50,51], in food and water intake [52], and in the regulation of autonomic responses [53]. Orexin neurons have a central role in the regulation of all these functions, and orexin autoreceptors have been proven to be part of a positive feedback loop [54,55]. The high expression of OX1R in these neurons in the present study suggests that this feedback loop also exists in the human brain, and it is possible that antagonists could intervene in this process, reducing orexin release.

OX1R is also highly expressed in the supraoptic and the paraventricular nuclei mainly, by the magnocellular neurons, which have complex dendritic trees. The dendrites of these neurons are very important sites for the synthesis and the release of vasopressin and oxytocin [56,57]. Studies on rodents have reported that OX1Rs are also found in magnocellular neurons of the rat hypothalamus [55], and that the administration of orexin to rats increases water intake [58]. Our findings imply that, in humans as well, orexin A may influence, through OX1R, the secretion of vasopressin and oxytocin and the regulation of drinking behavior. In the medial nuclei, OX1R is mainly expressed in the compact compartment of the dorsomedial nucleus, where neurons with a few, long dendrites are located. This nucleus, together with the ventromedial nucleus, participate in the regulation of food intake, metabolism, and autonomic drive [59,60,61]. Specifically, for the compact compartment of the dorsomedial nucleus, its function has been coupled with feeding-mediated regulation of circadian behaviors, forming direct connections with orexin neurons of the lateral hypothalamus [62]. It is therefore possible that, in the human brain, these functions are also related to the regulatory action of orexin A, which is the OX1R ligand.

Our study reveals a wide distribution of OX1R in the human hypothalamus. It shows that the ligands of this receptor, either orexin A or its agonists and antagonists, may influence diverse functions in humans related to metabolism, food and water intake, the sleep–wake cycle, autonomic responses, and reward. Thus, a very thorough testing of such pharmaceutical agents must be carried out, in order to strictly define patients with specific characteristics that could use these medications. Furthermore, the hypothalamic areas with strong OX1R immunoreactivity have large and complex neurons, with expanded dendritic trees. These findings could suggest that orexin A, which is the OX1R ligand, is important for the plasticity of these neurons, and for the general functioning of the human hypothalamic neural networks, in the human brain, as was recently reported in the rat brain [63]. The morphological and morphometric analysis of the neurons in the lateral hypothalamic area revealed that these neurons have specific morphological characteristics that make them ideal for use in neural network models, in order to study the functions of these nuclei [64]. This information is critical for understanding the underlying neural mechanisms that mediate the effects of the orexin system on brain function and behavior.

## Figures and Tables

**Figure 1 biomolecules-13-00592-f001:**
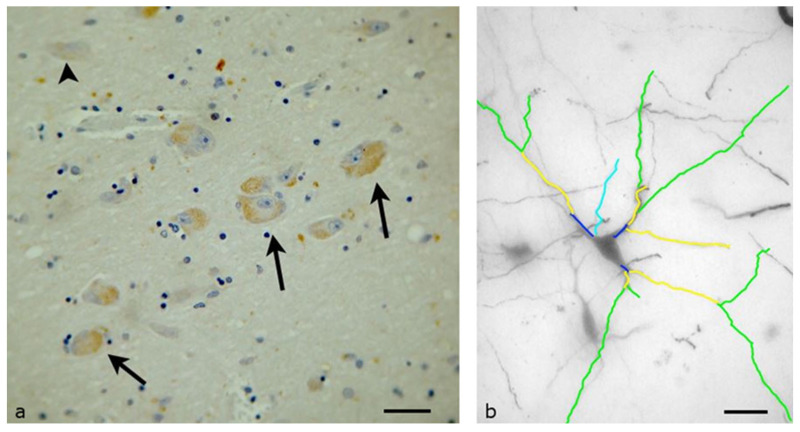
Lateral Hypothalamic Area of the human Hypothalamus: (**a**) Immunohistochemical staining, specific for OX1R. Small neural circuits of three to four neurons are more immunopositive (black arrow) than large, isolated neurons, which have little OX1R expression (arrow heads) (Scale bar, 25 μm); (**b**) A Golgi-stained neuron mapped by the ImageJ program. Primary (dark blue), secondary (yellow) and tertiary (green) dendrites can be found. Light blue depicts the axon of the neuron (Scale bar, 20 μm).

**Figure 2 biomolecules-13-00592-f002:**
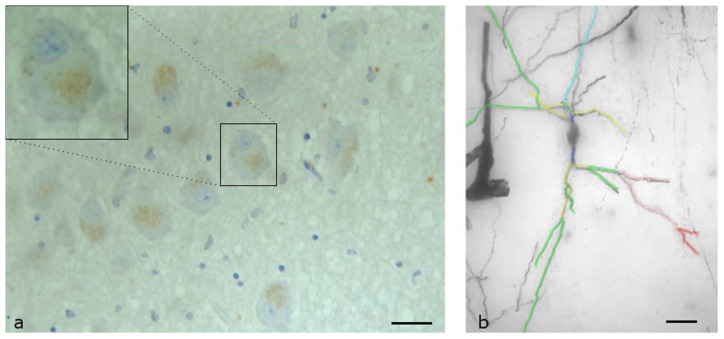
Lateral Preoptic Nucleus of human Hypothalamus: (**a**) Immunohistochemical staining, specific for OX1R. Almost all neurons (more than 95%) express OX1R but in lower quantities. The OX1Rs appear as cytoplasmic granules, clustered near the cell membrane (Scale bar, 25 μm); (**b**) A Golgi-stained neuron mapped by the ImageJ program. Primary (dark blue), secondary (yellow), tertiary (green), quaternary (pink) and even quinary (red) dendrites can be found. Light blue depicts the axon of the neuron (Scale bar, 30 μm).

**Figure 3 biomolecules-13-00592-f003:**
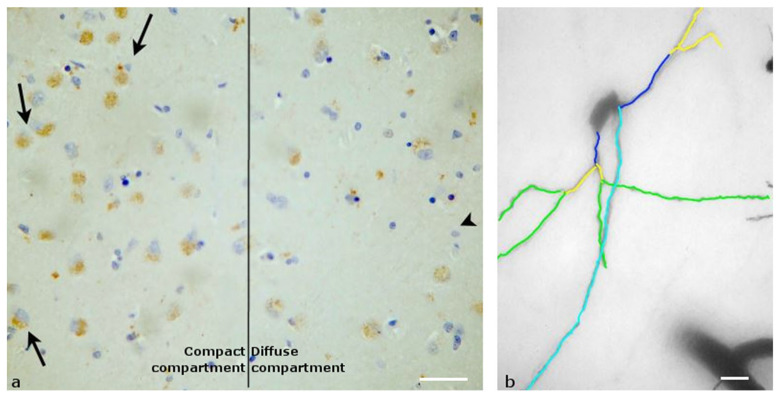
Dorsomedial nucleus of the human Hypothalamus: (**a**) Immunohistochemical staining, specific for OX1R. On the left of the photo is the compact compartment, with large neurons that are immunopositive (black arrow), and on the right of the photo is the diffuse compartment, with mainly smaller, non-immunopositive neurons (arrowhead) (Scale bar, 50 μm); (**b**) A Golgi-stained neuron of the compact compartment mapped by the ImageJ program. Primary (dark blue), secondary (yellow) and tertiary (green) dendrites can be found. Light blue depicts the axon of the neuron, which moves towards the lateral hypothalamic area (Scale bar, 10 μm).

**Figure 4 biomolecules-13-00592-f004:**
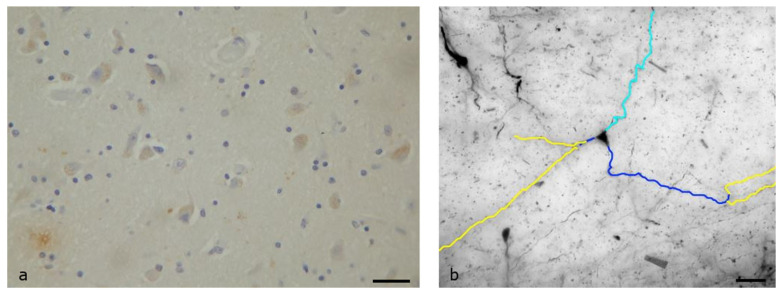
Ventromedial nucleus of the human Hypothalamus: (**a**) Immunohistochemical staining, specific for OX1R. More than 70% of both types of neurons were immunopositive in OX1R, but the stain was less intense for the smaller neurons (Scale bar, 30 μm); (**b**) A Golgi-stained neuron of the compact compartment, mapped by the ImageJ program. Only primary (dark blue) and secondary (yellow) dendrites can be found. Light blue depicts the axon of the neuron (Scale bar, 20 μm).

**Figure 5 biomolecules-13-00592-f005:**
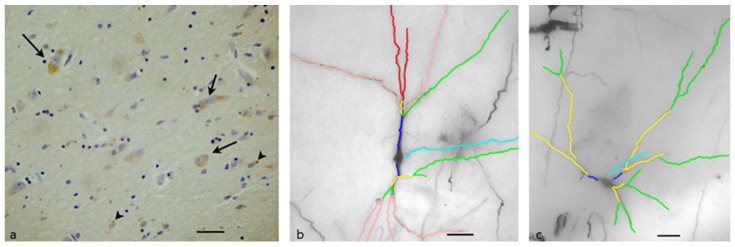
Supraoptic Nucleus of the human Hypothalamus: (**a**) Immunohistochemical staining specific for OX1R. SI neurons are more immunopositive (black arrow) than SII neurons, which have mild cytoplasmic staining, in the form of small deposits (arrow heads) (Scale bar, 30 μm); (**b**) A Golgi-stained SI neuron mapped by the ImageJ program. Primary (dark blue), secondary (yellow), tertiary (green), quaternary (pink) and even quinary (red) dendrites can be found. As such, SI neurons have some of the most complicated dendritic trees in the human hypothalamus. Light blue depicts the axon of the neuron (Scale bar, 30 μm); (**c**) A Golgi-stained SII neuron mapped by the ImageJ program. Primary (dark blue), secondary (yellow) and tertiary (green) dendrites can be found. Light blue depicts the axon of the neuron. SII neurons are smaller and have a simple dendritic tree (Scale bar, 20 μm).

**Figure 6 biomolecules-13-00592-f006:**
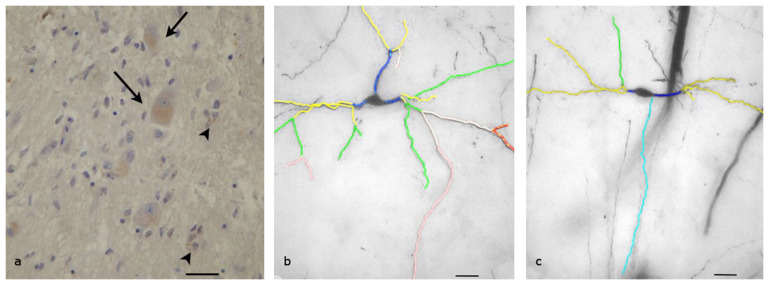
Paraventricular Nucleus of the human Hypothalamus: (**a**) Immunohistochemical staining, specific for OX1R. The big neurons, which are similar to SI neurons (black arrows), are more immunopositive for OX1R than the small neurons of the paraventricular nucleus (arrow heads) (Scale bar, 30 μm); (**b**) A large, Golgi-stained neuron mapped by the ImageJ program. Primary (dark blue), secondary (yellow), tertiary (green), quaternary (pink) and even quinary (red) dendrites can be found. Light blue depicts the axon of the neuron. The big neurons are similar to SI neurons of the supraoptic nucleus and can be found mainly in the magnocellular compartment (Scale bar, 30 μm); (**c**) A small Golgi-stained neuron mapped by the ImageJ program. Primary (dark blue), secondary (yellow) and tertiary (green) dendrites can be found. Light blue depicts the axon of the neuron. There is a variety of small neurons, which have many differences in morphology (Scale bar, 20 μm).

**Table 1 biomolecules-13-00592-t001:** Information of the individuals whose specimens were used in the study.

Patient	Gender	Age	Cause of Death
1	Male	66	Myocardial infarction
2	female	29	Traumatic brain injury—Car accident
3	Male	32	Respiratory infection
4	Male	86	Bolus aspiration
5	female	74	Myocardial infarction
6	Male	72	Pulmonary embolism
7	Male	20	Body injuries—Car accident
8	Male	80	Respiratory infection
9	Male	17	Traumatic brain injury—Car accident
10	female	51	Myocardial infarction
11	female	73	Pulmonary embolism
12	female	51	Respiratory infection
13	male	69	Myocardial infarction
14	male	23	Traumatic brain injury—Car accident
15	female	62	Rupture of ascending aorta

**Table 2 biomolecules-13-00592-t002:** Morphometric analysis of the neurons of the nuclei of the hypothalamus.

	Lateral Hypothalamic Area	Lateral Preoptic Nucleus	Dorsomedial Nucleus	Ventromedial Nucleus	Supraoptic Nucleus SI/SII Type	Paraventricular Nucleus
Number of Primary Dendrites (Average)	2–3 (2.5)	3–8 (4.8)	2–3 ^†^	2–3 (2.42)	2–5 (2.37)/2–4 (2.81)	2–3 ^†^
Range of length of Primary Dendrites (Average)	11.8–88.1 (39.4) μm	15.3–40.2 (30.1) μm	11.8–67.4 μm ^†^	46.2–126.3 (101.3) μm	16.4–49.1 (28.9)/16.1–79.4 (43.2) μm	10.7–93.6 μm ^†^
Range of total length of Primary dendrites (Average)	76.2–199.3 (96.7) μm	96.8–201.3 (138.4) μm	76.2–203.3 μm ^†^	100.3–370.6 (243.9) μm	32.8–123.7 (54.4)/64.4–312.5 (91.7) μm	42.8–263.7 μm ^†^
Number of Secondary Dendrites	3–8 (5.6)	3–9 (5.9)	3–8 ^†^	2–6 (3.02)	2–6 (3.46)/2–4 (2.9)	3–9 ^†^
Range of length of Secondary Dendrites (Average)	23.5–185.6 (107.1) μm	27.7–108.3 (69.8) μm	49.1–176.2 μm ^†^	123.5–172.1 (131.9) μm	14.7–71.3 (52.5)/36.2–102.3 (77.9) μm	12.4–159.6 μm ^†^
Range of total length of Secondary dendrites (Average)	330.9–951.9 (580.9) μm	305.6–912.9 (693.21) μm	302.4–951.1 μm ^†^	247.1–1008.2 (501.9) μm	88.2–369.6 (176.3)/72.4–396.9 (213.6) μm	137.9–1138.3 μm ^†^
Number of Tertiary Dendrites (Average)	1–9 ^†^	3–9 (6.46)	2–8 ^†^	0–2 ^†^	2–8 (4.21)/1–9 (4.65)	2–9 ^†^
Range of length of Tertiary Dendrites (Average)	28.4–264.1 μm ^†^	24.5–297.9 (156.3) μm	28.2–201.2 μm ^†^	0–108.7 μm ^†^	38.3–162.1 (96.8)/25.4–106.8 (64.2) μm	15.2–169.0 μm ^†^
Range of total length of Tertiary dendrites (Average)	102.3–878.3 μm ^†^	102.0–889.2 μm ^†^	100.2–731.0 μm ^†^	0–217.6 μm ^†^	285.6–843.2 (501.1)/58.6–489.8 (265.1) μm	65.1–972.8 μm ^†^
Number of Quaternary Dendrites (Average)	0–6 ^†^	0–5 ^†^	-	-	1–8 (4.67)/-	0–7 ^†^
Range of length of Quaternary Dendrites (Average)	0–185.0 μm ^†^	0–143.2 (64.9) μm	-	-	26.6–129.4 (76.2)/-μm	0–83.1 μm ^†^
Range of total length of Quaternary Dendrites (Average)	0–813.1 μm ^†^	0–468.5 μm ^†^	-	-	123.1–583.2 (352.7)/-μm	0–474.0 ^†^

-: No available data (dendrites were not found). †: No morphometric analysis (dendrites were variable or are found only in a low number of neurons).

## Data Availability

All data are freely available upon reasonable request. Please contact the corresponding author.

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
