# Peer review of "Cellular Localization of Orexin 1 Receptor in Human Hypothalamus and Morphological Analysis of Neurons Expressing the Receptor"

_biomolecules, 2023, doi:10.3390/biom13040592_

Round 1
Reviewer 1 Report
The paper presented for review is interesting due to the limited amount of research on the morphology of orexin receptor neurons. However, the paper has several weaknesses that require clarification by the authors before the study can be published.
1. Three patients died as a result of brain damage. Why was the hypothalamus of these patients included in the study? It is hard to imagine that the hypothalamus was not damaged as a result of traumatic brain injury. Therefore, the results may not be reliable.
2. Why are there no references to age groups and gender in the results and discussion sections? The orexin system undergoes significant morphological changes during the aging process. However, this topic has been omitted throughout the paper. Why?
The subject of differences in the morphology and functioning of orexin neurons and orexin receptors depending on sex is controversial and still remains unexplained. The research material consisted of brains taken from 6 women and 9 men, therefore, omitting the possible influence of gender on the research results reduces the value of the paper. Please explain why this topic has been omitted and include an appropriate correction in the paper.
3. The discussion is laconic. For example, it lacks a comparison of its own results with those derived from work on the morphology of OX1R neurons in other mammals besides rats. In addition, it is a pity that the work focused on examining the location of only the orexin 1 receptor in the hypothalamus.
4. Due to editorial remarks, a space before the µm is missing in many sentences.
The work requires major editing according to above comments. After corrections, it can be published in the journal Biomolecules.
Reviewer 2 Report
Konstantina Vraka et al. examined cellular localization of orexin 1 receptor in human hypothalamus and morphological analysis of neurons expressing the receptor.
I have listed my comments below:
1. The abstract doesn't contain a summary of the results as well as the principal conclusion of the study. I advise the authors emphasizing the importance of their findings also in the Discussion section
2. Has OX1R and OX2R mRNA expression been studied only in rats? Please state clearly in which species the expression of these receptors has been studied, especially in the hypothalamus.
3. What is the expression of orexin in the human brain?
4. The Materials and Methods section is very poor, so it is difficult to assess whether the study was well planned and performed. In addition, there is no data from the statistical analysis of the results. Thus, this section requires many explanations, e.g.:
- how many brains were dedicated to Nissl, Golgi and immunohistochemical staining? Which brains were dedicated to Nissl, Golgi and immunohistochemical staining? If all of them what was the protocol of fixation? Please confirm clearly the number of hypothalamic specimens used for particular method and which individuals (male, female, age) were selected to each of the method? Was tissue obtained from the same hypothalamus used for Nissl and IHC?
- how long were the brains fixed for Nissl and immunohistochemical staining?
- there is no image of the Nissl staining e.g. the location of the examined nuclei
- what was the thickness of the Nissl sections?
- please describe in detail how was the total number of neurons calculated? What was the number n? What was 100% of the population? How many neurons are in each of the studied nucleus? How many fields were analysed per specimen per nucleus? Has a statistical analysis of the data been performed? Results of the statistical analysis should be included.
- in Table 2, the authors present the data of the morphometric analysis of neurons of the hypothalamic nuclei without any statistics; the data should be supplemented with statistical analysis
- please describe the IHC protocol,
- what detection system was used?
- how did the authors check the specificity of the primary antibodies used?
- it would be nice to have representative IHC images (e.g. at 4x or 10x) to show the immunoreactivity of the whole nuclei studied
- please describe in the Material and Method section the Sholl analysis that was used in the study
5. Conclusions are highly speculative, conclusions should be based on the results
Round 2
Reviewer 1 Report
The authors made changes in the text in accordance with the review. I have no further comments.
Author Response
Thank you for the comments!
Reviewer 2 Report
Most of the comments were taken into account by the authors, however, I still have some concerns:
1. Authors claim that “This antibody is highly specific and have been used previously in a variety of immunohistochemical studies [34] – however they citied only one paper in addition on rats
2. Authors explain that “Each hypothalamus was initially separated into three different parts with coronal sections, anterior, middle, and posterior. From each part a thin specimen (coronal section of 0,5 mm thickness) was removed for Nissl staining and OX1R immunohistochemistry, while the rest was used for Golgi impregnation” Please explain how such a thin specimen (0.5 mm thick coronal section) was taken from each section. Manually? Technically it seems very difficult.
3. Authors claim that “Details about the calculation of the percentage of immunopositive neurons were added in line 330”. Where exactly these details were added, I couldn't find them. It is not explained at all how the total number of neurons was calculated.
4. According to the data presented in the Table 2, Authors emphasize that “Due to the small sample size and qualitative nature of the data analysis, further statistical analysis was not deemed although for example "Average length of Primary Dendrites" was presented. What does the authors mean by Average length, where is SD or SEM? Descriptive statistics should be added to the measurements.
Round 3
Reviewer 2 Report
As for the presentation of the results in Table 2 in connection with the limitations indicated by the authors, Authors should consider presenting these parameters in a different form, e.g. instead of the average length of the primary dendrites, provide the range of their length e.g. from ... to ...
